# Learning to See Physics via Visual De-animation

**Jiajun Wu**
MIT CSAIL

**Erika Lu**
University of Oxford

**Pushmeet Kohli**
DeepMind

**William T. Freeman**
MIT CSAIL, Google Research

**Joshua B. Tenenbaum**
MIT CSAIL

## Abstract

We introduce a paradigm for understanding physical scenes without human annotations. At the core of our system is a physical world representation that is first recovered by a perception module and then utilized by physics and graphics engines. During training, the perception module and the generative models learn by *visual de-animation* — interpreting and reconstructing the visual information stream. During testing, the system first recovers the physical world state, and then uses the generative models for reasoning and future prediction.

Even more so than forward simulation, inverting a physics or graphics engine is a computationally hard problem; we overcome this challenge by using a convolutional inversion network. Our system quickly recognizes the physical world state from appearance and motion cues, and has the flexibility to incorporate both differentiable and non-differentiable physics and graphics engines. We evaluate our system on both synthetic and real datasets involving multiple physical scenes, and demonstrate that our system performs well on both physical state estimation and reasoning problems. We further show that the knowledge learned on the synthetic dataset generalizes to constrained real images.

## 1 Introduction

Inspired by human abilities, we wish to develop machine systems that understand scenes. Scene understanding has multiple defining characteristics which break down broadly into two features. First, human scene understanding is *rich*. Scene understanding is physical, predictive, and causal: rather than simply knowing what is where, one can also predict what may happen next, or what actions one can take, based on the physics afforded by the objects, their properties, and relations. These predictions, hypotheticals, and counterfactuals are probabilistic, integrating uncertainty as to what is more or less likely to occur. Second, human scene understanding is *fast*. Most of the computation has to happen in a single, feedforward, bottom-up pass.

There have been many systems proposed recently to tackle these challenges, but existing systems have architectural features that allow them to address one of these features but not the other. Typical approaches based on inverting graphics engines and physics simulators [Kulkarni et al., 2015b] achieve richness at the expense of speed. Conversely, neural networks such as PhysNet [Lerer et al., 2016] are fast, but their ability to generalize to rich physical predictions is limited.

We propose a new approach to combine the best of both. Our overall framework for representation is based on graphics and physics engines, where graphics is run in reverse to build the initial physical scene representation, and physics is then run forward to imagine what will happen next or what can be done. Graphics can also be run in the forward direction to visualize the outputs of the physics simulation as images of what we expect to see in the future, or under different viewing conditions. Rather than use traditional, often slow inverse graphics methods [Kulkarni et al., 2015b], we learn to

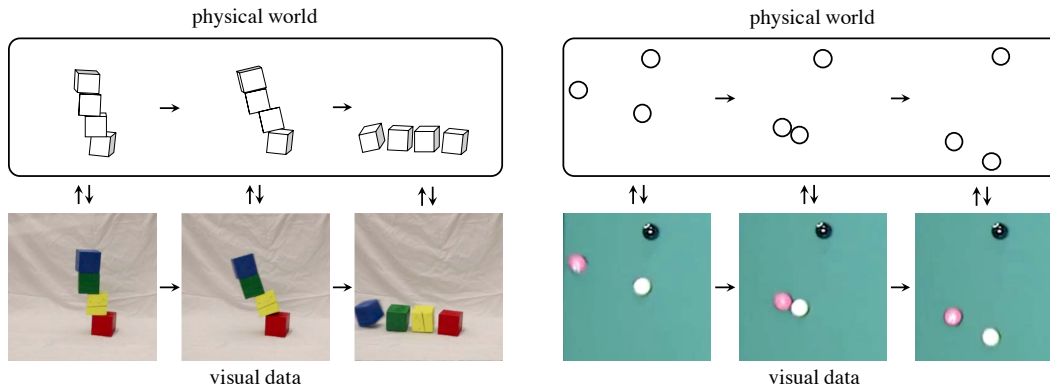

Figure 1: Visual de-animation — we would like to recover the physical world representation behind the visual input, and combine it with generative physics simulation and rendering engines.

invert the graphics engine efficiently using convolutional nets. Specifically, we use deep learning to train recognition models on the objects in our world for object detection, structure and viewpoint estimation, and physical property estimation. Bootstrapping from these predictions, we then infer the remaining scene properties through inference via forward simulation of the physics engine.

Without human supervision, our system learns by *visual de-animation*: interpreting and reconstructing visual input. We show the problem formulation in Figure 1. The simulation and rendering engines in the framework force the perception module to extract physical world states that best explain the data. As the physical world states are inputs to physics and graphics engines, we simultaneously obtain an interpretable, disentangled, and compact physical scene representation.

Our framework is flexible and adaptable to a number of graphics and physics engines. We present model variants that use neural, differentiable physics engines [Chang et al., 2017], and variants that use traditional physics engines, which are more mature but non-differentiable [Coumans, 2010]. We also explore various graphics engines operating at different levels, ranging from mid-level cues such as object velocity, to pixel-level rendering of images.

We demonstrate our system on real and synthetic datasets across multiple domains: synthetic billiard videos [Fragkiadaki et al., 2016], in which balls have varied physical properties, real billiard videos from the web, and real images of block towers from Facebook AI Research [Lerer et al., 2016].

Our contributions are three-fold. First, we propose a novel generative pipeline for physical scene understanding, and demonstrate its flexibility by incorporating various graphics and physics engines. Second, we introduce the problem of visual de-animation – learning rich scene representations without supervision by interpreting and reconstructing visual input. Third, we show that our system performs well across multiple scenarios and on both synthetic and constrained real videos.

## 2   Related Work

Physical scene understanding has attracted increasing attention in recent years [Gupta et al., 2010, Jia et al., 2015, Lerer et al., 2016, Zheng et al., 2015, Battaglia et al., 2013, Mottaghi et al., 2016b, Fragkiadaki et al., 2016, Battaglia et al., 2016, Mottaghi et al., 2016a, Chang et al., 2017, Agrawal et al., 2016, Pinto et al., 2016, Finn et al., 2016, Hamrick et al., 2017, Ehrhardt et al., 2017, Shao et al., 2014, Zhang et al., 2016]. Researchers have attempted to go beyond the traditional goals of high-level computer vision, inferring "what is where", to capture the physics needed to predict the immediate future of dynamic scenes, and to infer the actions an agent should take to achieve a goal. Most of these efforts do not attempt to learn physical object representations from raw observations. Some systems emphasize learning from pixels but without an explicitly object-based representation [Lerer et al., 2016, Mottaghi et al., 2016b, Fragkiadaki et al., 2016, Agrawal et al., 2016, Pinto et al., 2016, Li et al., 2017], which makes generalization challenging. Others learn a flexible model of the dynamics of object interactions, but assume a decomposition of the scene into physical objects and their properties rather than learning directly from images [Chang et al., 2017, Battaglia et al., 2016].

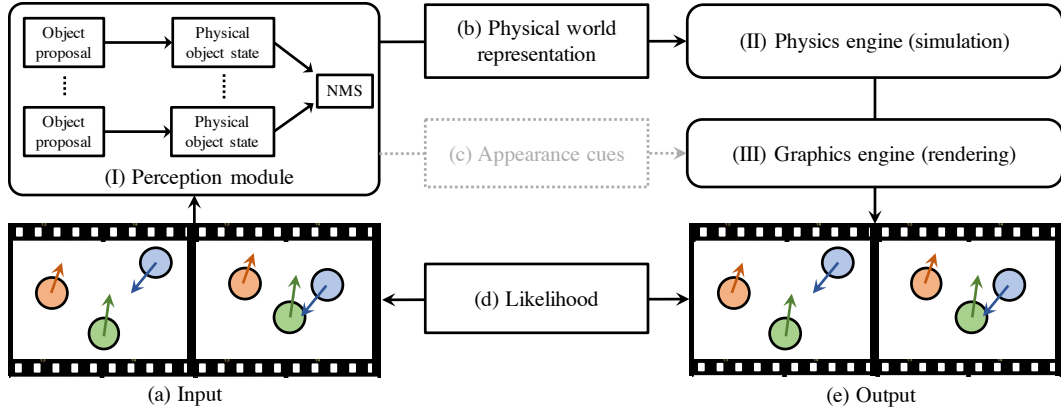

Figure 2: Our visual de-animation (VDA) model contains three major components: a convolutional perception module (I), a physics engine (II), and a graphics engine (III). The perception module efficiently inverts the graphics engine by inferring the physical object state for each segment proposal in input (a), and combines them to obtain a physical world representation (b). The generative physics and graphics engines then run forward to reconstruct the visual data (e). See Section 3 for details.

There have been some works that aim to estimate physical object properties [Wu et al., 2016, 2015, Denil et al., 2017]. Wu et al. [2015] explored an analysis-by-synthesis approach that is easily generalizable, but less efficient. Their framework also lacked a perception module. Denil et al. [2017] instead proposed a reinforcement learning approach. These approaches, however, assumed strong priors of the scene, and approximate object shapes with primitives. Wu et al. [2016] used a feed-forward network for physical property estimation without assuming prior knowledge of the environment, but the constrained setup did not allow interactions between multiple objects. By incorporating physics and graphics engines, our approach can jointly learn the perception module and physical model, optionally in a Helmholtz machine style [Hinton et al., 1995], and recover an explicit physical object representation in a range of scenarios.

Another line of related work is on future state prediction in either image pixels [Xue et al., 2016, Mathieu et al., 2016] or object trajectories [Kitani et al., 2017, Walker et al., 2015]. There has also been abundant research making use of physical models for human or scene tracking [Salzmann and Urtasun, 2011, Kyriazis and Argyros, 2013, Vondrak et al., 2013, Brubaker et al., 2009]. Our model builds upon and extends these ideas by jointly modeling an approximate physics engine and a perceptual module, with wide applications including, but not limited to, future prediction.

Our framework also relates to the field of "vision as inverse graphics" [Zhu and Mumford, 2007, Yuille and Kersten, 2006, Bai et al., 2012]. Connected to but different from traditional analysis-by-synthesis approaches, recent works explored using deep neural networks to efficiently explain an object [Kulkarni et al., 2015a, Rezende et al., 2016], or a scene with multiple objects [Ba et al., 2015, Huang and Murphy, 2015, Eslami et al., 2016]. In particular, Wu et al. [2017] proposed "scene de-rendering", building an object-based, structured representation from a static image. Our work incorporates inverse graphics with simulation engines for physical scene understanding and scene dynamics modeling.

## 3 Visual De-animation

Our visual de-animation (VDA) model consists of an efficient inverse graphics component to build the initial physical world representation from visual input, a physics engine for physical reasoning of the scene, and a graphics engine for rendering videos. We show the framework in Figure 2. In this section, we first present an overview of the system, and then describe each component in detail.

### 3.1 Overview

The first component of our system is an approximate inverse graphics module for physical object and scene understanding, as shown in Figure 2-I. Specifically, the system sequentially computes object proposals, recognizes objects and estimates their physical state, and recovers the scene layout.

The second component of our system is a physics engine, which uses the physical scene representation recovered by the inverse graphics module to simulate future dynamics of the environment (Figure 2-II). Our system adapts to both neural, differentiable simulators, which can be jointly trained with the perception module, and rigid-body, non-differentiable simulators, which can be incorporated using methods such as REINFORCE [Williams, 1992].

The third component of our framework is a graphics engine (Figure 2-III), which takes the scene representations from the physics engine and re-renders the video at various levels (*e.g.* optical flow, raw pixel). The graphics engine may need additional appearance cues such as object shape or color (Figure 2c). Here, we approximate them using simple heuristics, as they are not a focus of our paper. There is a tradeoff between various rendering levels: while pixel-level reconstruction captures details of the scene, rendering at a more abstract level (*e.g.* silhouettes) may better generalize. We then use a likelihood function (Figure 2d) to evaluate the difference between synthesized and observed signals, and compute gradients or rewards for differentiable and non-differentiable systems, respectively.

Our model combines efficient and powerful deep networks for recognition with rich simulation engines for forward prediction. This provides us two major advantages over existing methods: first, simulation engines take an interpretable representation of the physical world, and can thus easily generalize and supply rich physical predictions; second, the model learns by explaining the observations — it can be trained in a self-supervised manner without requiring human annotations.

## 3.2 Physical Object and Scene Modeling

We now discuss each component in detail, starting with the perception module.

**Object proposal generation** Given one or a few frames (Figure 2a), we first generate a number of object proposals. The masked images are then used as input to the following stages of the pipeline.

**Physical object state estimation** For each segment proposal, we use a convolutional network to recognize the physical state of the object, which consists of intrinsic properties such as shape, mass, and friction, as well as extrinsic properties such as 3D position and pose. The input to the network is the masked image of the proposal, and the output is an interpretable vector for its physical state.

**Physical world reconstruction** Given objects' physical states, we first apply non-maximum suppression to remove object duplicates, and then reconstruct the physical world according to object states. The physical world representation (Figure 2b) will be employed by the physics and graphics engines for simulation and rendering.

## 3.3 Physical Simulation and Prediction

The two types of physics engines we explore in this paper include a neural, differentiable physics engine and a standard rigid-body simulation engine.

**Neural physics engines** The neural physics engine is an extension of the recent work from Chang et al. [2017], which simulates scene dynamics by taking object mass, position, and velocity. We extend their framework to model object friction in our experiments on billiard table videos. Though basic, the neural physics engine is differentiable, and thus can be end-to-end trained with our perception module to explain videos. Please refer to Chang et al. [2017] for details of the neural physics engine.

**Rigid body simulation engines** There exist rather mature, rigid-body physics simulation engines, *e.g.* Bullet [Coumans, 2010]. Such physics engines are much more powerful, but non-differentiable. In our experiments on block towers, we used a non-differentiable simulator with multi-sample REINFORCE [Rezende et al., 2016, Mnih and Rezende, 2016] for joint training.

## 3.4 Re-rendering with a Graphics Engine

In this work, we consider two graphics engines operating at different levels: for the billiard table scenario, we use a renderer that takes the output of a physics engine and generates pixel-level rendering; for block towers, we use one that computes only object silhouettes.

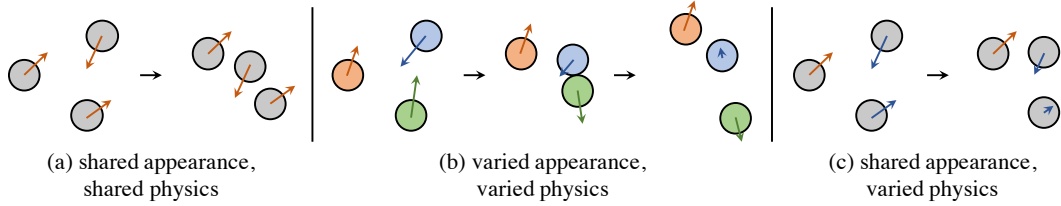

(a) shared appearance,
shared physics

(b) varied appearance,
varied physics

(c) shared appearance,
varied physics

Figure 3: The three settings of our synthetic billiard videos: (a) balls have the same appearance and physical properties, where the system learns to discover them and simulate the dynamics; (b) balls have the same appearance but different physics, and the system learns their physics from motion; (c) balls have varied appearance and physics, and the system learns to associate appearance cues with underlying object states, even from a single image.

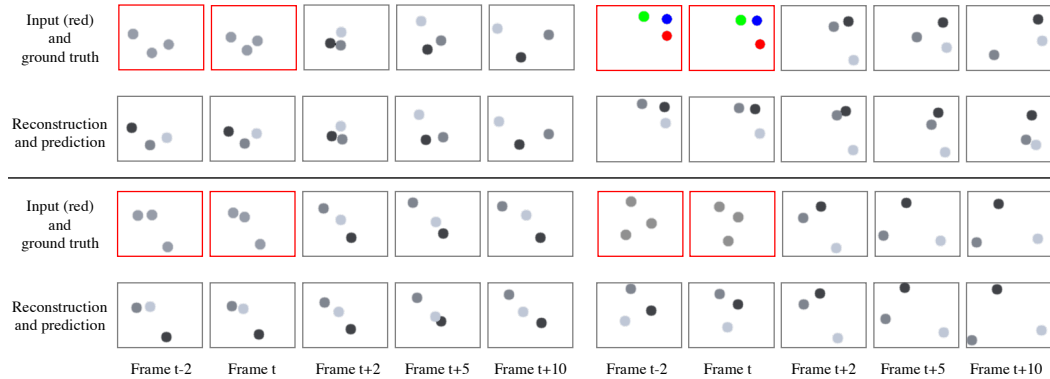

Figure 4: Results on the billiard videos, comparing ground truth videos with our predictions. We show two of three input frames (in red) due to space constraints. Left: balls share appearance and physics (I), where our framework learns to discover objects and simulate scene dynamics. Top right: balls have different appearance and physics (II), where our model learns to associate appearance with physics and simulate collisions. It learns that the green ball should move further than the heavier blue ball after the collision. Bottom right: balls share appearance but have different frictions (III), where our model learns to associate motion with friction. It realizes from three input frames that the right-most ball in the first frame has a large friction coefficient and will stop before the other balls.

# 4 Evaluation

We evaluate variants of our frameworks in three scenarios: synthetic billiard videos, real billiard videos, and block towers. We also test how models trained on synthetic data generalize to real cases.

## 4.1 Billiard Tables: A Motivating Example

We begin with synthetic billiard videos to explore end-to-end learning of the perceptual module along with differentiable simulation engines. We explore how our framework learns the physical object state (position, velocity, mass, and friction) from its appearance and/or motion.

**Data**   For the billiard table scenario, we generate data using the released code from Fragkiadaki et al. [2016]. We updated the code to allow balls of different mass and friction. We used the billiard table scenario as an initial exploration of whether our models can learn to associate visual object appearance and motion with physical properties. As shown in Figure 3, we generated three subsets, in which balls may have shared or differing appearance (color), and physical properties. For each case, we generated 9,000 videos for training and 200 for testing.

*(I) Shared appearance and physics (Figure 3a):* balls all have the same appearance and the same physical properties. This basic setup evaluates whether we can jointly learn an object (ball) discoverer and a physics engine for scene dynamics.

| Datasets | | Methods | Recon. | Position Prediction (Abs) | | | | Velocity Prediction (Abs) | | | |
|---|---|---|---|---|---|---|---|---|---|---|---|
| Appear. | Physics | | Pixel MSE | 1st | 5th | 10th | 20th | 1st | 5th | 10th | 20th |
| Same | Same | Baseline | 0.046 | 4.58 | 18.20 | 46.06 | 119.97 | 2.95 | 5.63 | 7.32 | 8.45 |
| | | VDA (init) | 0.046 | 3.46 | 6.61 | 12.76 | 26.10 | 1.41 | 1.97 | 2.34 | 2.65 |
| | | VDA (full) | 0.044 | 3.25 | 6.52 | 12.34 | 25.55 | 1.37 | 1.87 | 2.22 | 2.55 |
| Diff | Diff. | Baseline | 0.058 | 6.57 | 26.38 | 70.47 | 180.04 | 3.78 | 7.62 | 10.51 | 12.02 |
| | | VDA (init) | 0.058 | 3.82 | 8.92 | 17.09 | 34.65 | 1.65 | 2.27 | 3.02 | 3.21 |
| | | VDA (full) | 0.055 | 3.55 | 8.58 | 16.33 | 32.97 | 1.64 | 2.20 | 2.89 | 3.05 |
| Same | Diff. | Baseline | 0.038 | 9.58 | 79.78 | 143.67 | 202.56 | 12.37 | 23.42 | 25.08 | 23.98 |
| | | VDA (init) | 0.038 | 6.06 | 19.75 | 34.24 | 46.40 | 3.37 | 5.16 | 5.01 | 3.77 |
| | | VDA (full) | 0.035 | 5.77 | 19.34 | 33.25 | 43.42 | 3.23 | 4.98 | 4.77 | 3.35 |

Table 1: Quantitative results on synthetic billiard table videos. We evaluate our visual de-animation (VDA) model before and after joint training (init vs. full). For scene reconstruction, we compute MSE between rendered images and ground truth. For future prediction, we compute the Manhattan distance in pixels between predicted object position and velocity and ground truth in pixels, at the 1st, 5th, 10th, and 20th future frames. Our full model performs better. See qualitative results in Figure 4.

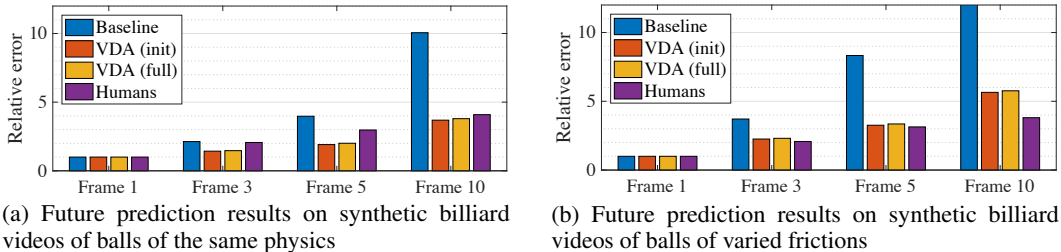

(a) Future prediction results on synthetic billiard videos of balls of the same physics

(b) Future prediction results on synthetic billiard videos of balls of varied frictions

Figure 5: Behavioral study results on future position prediction of billiard videos where balls have the same physical properties (a), and balls have varied physical properties (b). For each model and humans, we compare how their long-term relative prediction error grows, by measuring the ratio with respect to errors in predicting the first next frame. Compared to the baseline model, the behavior of our prediction models aligns well with human predictions.

*(II) Varied appearance and physics (Figure 3b):* balls can be of three different masses (light, medium, heavy), and two different friction coefficients. Each of the six possible combinations is associated with a unique color (appearance). In this setup, the scene de-rendering component should be able to associate object appearance with its physical properties, even from a single image.

*(III) Shared appearance, varied physics (Figure 3c):* balls have the same appearance, but have one of two different friction coefficients. Here, the perceptual component should be able to associate object motion with its corresponding friction coefficients, from just a few input images.

**Setup** For this task, the physical state of an object is its intrinsic properties, including mass $m$ and friction $f$, and its extrinsic properties, including 2D position $\{x, y\}$ and velocity $v$. Our system takes three 256×256 RGB frames $I_1, I_2, I_3$ as input. It first obtains flow fields from $I_1$ to $I_2$ and from $I_2$ to $I_3$ by a pretrained spatial pyramid network (SPyNet) [Ranjan and Black, 2017]. It then generates object proposals by applying color filters on input images.

Our perceptual model is a ResNet-18 [He et al., 2015], which takes as input three masked RGB frames and two masked flow images of each object proposal, and recovers the object's physical state. We use a differentiable, neural physics engine with object intrinsic properties as parameters; at each step, it predicts objects' extrinsic properties (position $\{x, y\}$ and velocity $v$) in the next frame, based on their current estimates. We employ a graphics engine that renders original images from the predicted positions, where the color of the balls is set as the mean color of the input object proposal. The likelihood function compares, at a pixel level, these rendered images and observations. It is straightforward to compute the gradient of object position from rendered RGB images and ground truth. Thus, this simple graphics engine is also differentiable, making our system end-to-end trainable.

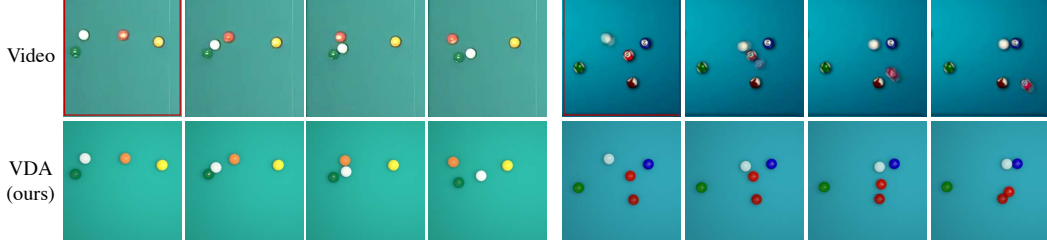

Figure 6: Sample results on web videos of real billiard games and computer games with realistic rendering. Left: our method correctly estimates the trajectories of multiple objects. Right: our framework correctly predicts the two collisions (white vs. red, white vs. blue), despite the motion blur in the input, though it underestimates the velocity of the red ball after the collision. Note that the billiard table is a chaotic system, and highly accurate long-term prediction is intractable.

Our training paradigm consists of two steps. First, we pretrain the perception module and the neural physics engine separately on synthetic data, where ground truth is available. The second step is end-to-end fine-tuning without annotations. We observe that the framework does not converge well without pre-training, possibly due to the multiple hypotheses that can explain a scene (*e.g.*, we can only observe relative, not absolute masses from collisions). We train our framework using SGD, with a learning rate of 0.001 and a momentum of 0.9. We implement our framework in Torch7 [Collobert et al., 2011]. During testing, the perception module is run in reverse to recover object physical states, and the learned physics engine is then run in forward for future prediction.

**Results**    Our formulation recovers a rich representation of the scene. With the generative models, we show results in scene reconstruction and future prediction. We compare two variants of our algorithm: the initial system has its perception module and neural physics engine separately trained, while the full system has an additional end-to-end fine-tuning step, as discussed above. We also compare with a baseline, which has the sample perception model, but in prediction, simply repeats object dynamics in the past without considering interactions among them.

*Scene reconstruction:* given input frames, we are able to reconstruct the images based on inferred physical states. For evaluation, we compute pixel-level MSE between reconstructions and observed images. We show qualitative results in Figure 4 and quantitative results in Table 1.

*Future prediction:* with the learned neural simulation engine, our system is able to predict future events based on physical world states. We show qualitative results in Figure 4 and quantitative results in Table 1, where we compute the Manhattan distance in pixels between predicted positions and velocities and the ground truth. Our model achieves good performance in reconstructing the scene, understanding object physics, and predicting scene dynamics. See caption for details.

**Behavioral studies**    We further conduct behavioral studies, where we select 50 test cases, show the first three frames of each to human subjects, and ask them the positions of each ball in future frames. We test 3 subjects per case on Amazon Mechanical Turk. For each model and humans, we compare how their long-term relative prediction error grows, by measuring the ratio with respect to errors in predicting the first next frame. As shown in Figure 5, the behavior of our models resembles human predictions much better than the baseline model.

## 4.2    Billiard Tables: Transferring to Real Videos

**Data**    We also collected videos from YouTube, segmenting them into two-second clips. Some videos are from real billiard competitions, and the others are from computer games with realistic rendering. We use it as an out-of-sample test set for evaluating the model's generalization ability.

**Setup and Results**    Our setup is the same as that in Section 4.1, except that we now re-train the perceptual model on the synthetic data of varied physics, but with flow images as input instead of RGB images. Flow images abstract away appearance changes (color, lighting, *etc*.), allowing the model to generalize better to real data. We show qualitative results of reconstruction and future prediction in Figure 6 by rendering our inferred representation using the graphics software, Blender.

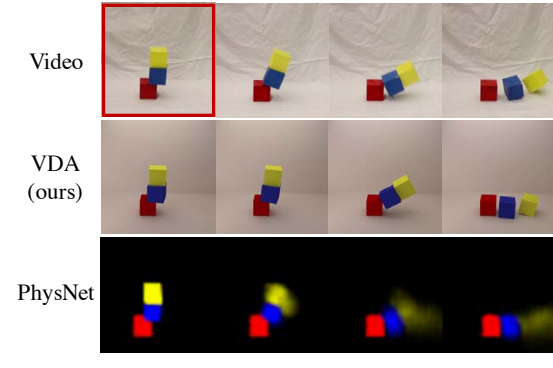

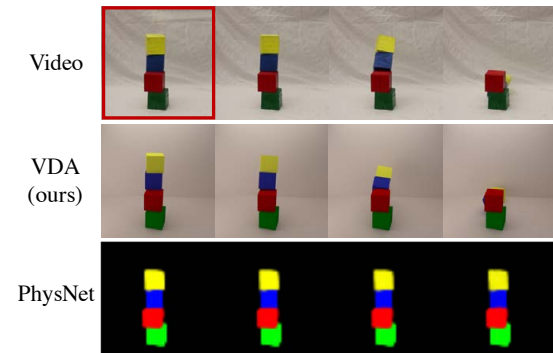

| Methods | # Blocks | | | Mean |
|---|---|---|---|---|
| | 2 | 3 | 4 | |
| Chance | 50 | 50 | 50 | 50 |
| Humans | 67 | 62 | 62 | 64 |
| PhysNet | 66 | 66 | 73 | 68 |
| GoogLeNet | 70 | 70 | 70 | 70 |
| VDA (init) | 73 | 74 | 72 | 73 |
| VDA (joint) | 75 | 76 | 73 | 75 |
| VDA (full) | 76 | 76 | 74 | 75 |

(b) Accuracy (%) of stability prediction on the blocks dataset

| Methods | 2 | 3 | 4 | Mean |
|---|---|---|---|---|
| PhysNet | 56 | 68 | 70 | 65 |
| GoogLeNet | 70 | 67 | 71 | 69 |
| VDA (init) | 74 | 74 | 67 | 72 |
| VDA (joint) | 75 | 77 | 70 | 74 |
| VDA (full) | 76 | 76 | 72 | 75 |

(c) Accuracy (%) of stability prediction when trained on synthetic towers of 2 and 4 blocks, and tested on all block tower sizes.

(a) Our reconstruction and prediction results given a single frame (marked in red). From top to bottom: ground truth, our results, results from Lerer et al. [2016].

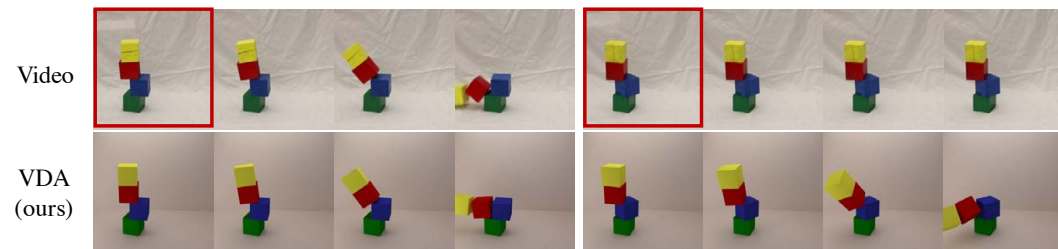

(d) Our reconstruction and prediction results given a single frame (marked in red)

Figure 7: Results on the blocks dataset [Lerer et al., 2016]. For quantitative results (b), we compare three variants of our visual de-animation (VDA) model: perceptual module trained without fine-tuning (init), joint fine-tuning with REINFORCE (joint), and full model considering stability constraint (full). We also compare with PhysNet [Lerer et al., 2016] and GoogLeNet [Szegedy et al., 2015].

## 4.3 The Blocks World

We now look into a different scenario — block towers. In this experiment, we demonstrate the applicability of our model to explain and reason from a static image, instead of a video. We focus on the reasoning of object states in the 3D world, instead of physical properties such as mass. We also explore how our framework performs with non-differentiable simulation engines, and how physics signals (*e.g.*, stability) could help in physical reasoning, even when given only a static image.

**Data** Lerer et al. [2016] built a dataset of 492 images of real block towers, with ground truth stability values. Each image may contain 2, 3, or 4 blocks of red, blue, yellow, or green color. Though the blocks are the same size, their sizes in each 2D image differ due to 3D-to-2D perspective transformation. Objects are made of the same material and thus have identical mass and friction.

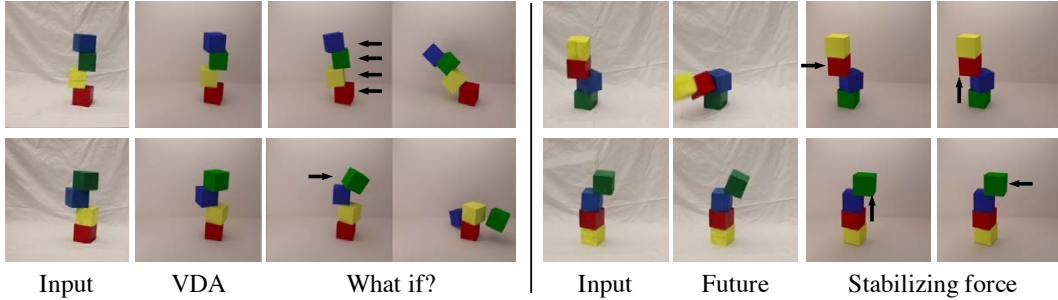

| Input | VDA | What if? | Input | Future | Stabilizing force |

Figure 8: Examples of predicting hypothetical scenarios and actively engaging with the scene. Left: predictions of the outcome of forces applied to two stable towers. Right: multiple ways to stabilize two unstable towers.

**Setup** Here, the physical state of an object (block) consists of its 3D position $\{x, y, z\}$ and 3D rotation (roll, pitch, yaw, each quantized into 20 bins). Our perceptual model is again a ResNet-18 [He et al., 2015], which takes block silhouettes generated by simple color filters as input, and recovers the object's physical state. For this task, we implement an efficient, non-differentiable, rigid body simulator, to predict whether the blocks are stable. We also implement a graphics engine to render object silhouettes for reconstructing the input. Our likelihood function consists of two terms: MSE between rendered silhouettes and observations, and the binary cross-entropy between the predicted stability and the ground truth stability.

Our training paradigm resembles the classic wake-sleep algorithm [Hinton et al., 1995]: first, generate 10,000 training images using the simulation engines; second, train the perception module on synthetic data with ground truth physical states; third, end-to-end fine-tuning of the perceptual module by explaining an additional 100,000 synthetic images without annotations of physical states, but with binary annotations of stability. We use multi-sample REINFORCE [Rezende et al., 2016, Mnih and Rezende, 2016] with 16 samples per input, assuming each position parameter is from a Gaussian distribution and each rotation parameter is from a multinomial distribution (quantized into 20 bins). We observe that the training paradigm helps the framework converge. The other setting is the same as that in Section 4.1.

**Results** We show results on two tasks: scene reconstruction and stability prediction. For each task, we compare three variants of our algorithm: the initial system has its perception module trained without fine-tuning; an intermediate system has joint end-to-end fine-tuning, but without considering the physics constraint; and the full system considers both reconstruction and physical stability during fine-tuning.

We show qualitative results on scene reconstruction in Figures 7a and 7d, where we also demonstrate future prediction results by exporting our inferred physical states into Blender. We show quantitative results on stability prediction in Table 7b, where we compare our models with PhysNet [Lerer et al., 2016] and GoogleNet [Szegedy et al., 2015]. All given a static image as test input, our algorithms achieve higher prediction accuracy (75% vs. 70%) efficiently (<10 milliseconds per image).

Our framework also generalizes well. We test out-of-sample generalization ability, where we train our model on 2- and 4-block towers, but test it on all tower sizes. We show results in Table 7c. Further, in Figure 8, we show examples where our physical scene representation combined with a physics engine can easily make conditional predictions, answering "What happens if..."-type questions. Specifically, we show frame prediction of external forces on stable block towers, as well as ways that an agent can stabilize currently unstable towers, with the help of rich simulation engines.

## 5   Discussion

We propose combining efficient, bottom-up, neural perception modules with rich, generalizable simulation engines for physical scene understanding. Our framework is flexible and can incorporate various graphics and physics engines. It performs well across multiple synthetic and real scenarios, reconstructing the scene and making future predictions accurately and efficiently. We expect our framework to have wider applications in the future, due to the rapid development of scene description languages, 3D reconstruction methods, simulation engines and virtual environments.

**Acknowledgements**

We thank Michael Chang, Donglai Wei, and Joseph Lim for helpful discussions. This work is supported by NSF #1212849 and #1447476, ONR MURI N00014-16-1-2007, the Center for Brain, Minds and Machines (NSF #1231216), Toyota Research Institute, Samsung, Shell, and the MIT Advanced Undergraduate Research Opportunities Program (SuperUROP).

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
