[Reviews · NeurIPS 2017]

Reviewer 1



This paper presents an approach to physical scene understanding which utilizes the combination of physics engines and renderers. The approach trains a network to predict a scene representation based on a set of input frames and subject to the constraints of the physics engine. This is trained using either SGD or reinforcement learning depending on whether a differentiable physics engine is used. The approach is applied to several simple scenarios with both real and synthetic data. Overall the idea in the paper is interesting and promising, however, the paper as it's written has several issues. Some of these may be able to be addressed in a rebuttal. First, there are some significant missing references, specifically related to using physical models in human tracking. Kyriazis and Argyros (CVPR2013, CVPR 2014); Pham et al (PAMI 2016); Brubaker et al (ICCV 2009, IJCV 2010); Vondrak et al (SIGGRAPH 2012, PAMI 2013). In terms of simple billiards-like scenarios, see also Salzmann and Urtasun (ICCV 2011). This is a significant blind spot in the related works and should be addressed. Second, the paper is missing some important technical details. In particular, it is never specified how the system actually works at test time. Is it simply run on successive sets of three input frames independently using a forward pass of the perception model? Does this mean that the physics and graphics rendering engines are not used at all at test time? If this is the case, why not use the physics engine at test time? It could, for instance, be used as a smoothing prior on the per-frame predictions of the perception model? This issue is particularly relevant for prediction of stability in the block world scenario. Is the network only run on a single set of three frames? This makes it particularly difficult to assess the technical aspects of this work. Other comments: - Absolute mass is never predictable without knowing something about what an object is made of. Only relative mass plays a role in the physics and visual perception can only tell you volume (assuming calibration), not density. Any network which appears to "learn" to predict absolute mass is really just learning biases about mass/density from the training set. This is fine, but important to recognize so that overboard claims aren't made. - The table entries and captions are inconsistent in Fig 7(b-c). The table used "full" and "full+", whereas the caption references "joint" and "full".

Reviewer 2



SUMMARY In this paper, the authors presented a data-driven approach for physical scene understanding: given a scene image, the system detects objects with state estimation, then reconstructs the scene, simulates it in a physics engine to predict its future states and render the prediction back to scene images. The approach has been evaluated on two tasks, namely the billiard game and the block stability prediction, and has shown better performance over existing methods. STRENGTH The most prominent contribution is the paper is probably among the first to construct a pipeline to fully reconstruct both a physical simulation and graphics reconstruction for physics scene understanding. Also, the method has been proved to work on the provided tasks in paritcular, the end-to-end training did prove to outperform separately training of individual components. Further, the description is general clear albeit it's on a very coarse level which is detailed as follows. WEAKNESS AND CONCONERN 1) As already mentioned in previous section, the description lacks certain levels of details for complete reproduction, for example, how is the physics engine implemented, it's understandable that the authors left out some details with proper reference, however it is not very clear, in paricular as physics engine is an important component in the system, how the engine is set up and the set up affect the results. 2) In addition to 2), there are concerns about the evaluation protocols for the billiard cases. why didn't the authors compared the results to previous results on the datasets other than the proposed baseline (sample perception model + repeating object dynamics) in the paper; for the block stability prediction, are the settings comparable to ones in the previous results. Those are important details to shed more lights to see if the proposed fully reconstruction and simulation approach did make a differences on a specific task over existing method, in particular the end-to-end without reconstruction as in the block stability prediction, though the authors can argue that full construction may be easier for rendering more articulated prediction results. 3) The authors should also cite recent work on frame prediction. this is very related.